# Reviewing the UK's Action Levels for the Management of Dredged Material

**Claire Mason** [1,*], **Chris Vivian** [2], **Andrew Griffith** [1], **Lee Warford** [1], **Clare Hynes** [1], **Jon Barber** [1], **David Sheahan** [1], **Philippe Bersuder** [1], **Adil Bakir** [1] and **Jemma-Anne Lonsdale** [1,*]

1    Centre for Environment, Fisheries and Aquaculture Science, Pakefield Road, Lowestoft NR33 0HT, UK; andrew.griffith@cefas.co.uk (A.G.); lee.warford@cefas.co.uk (L.W.); clare.hynes@cefas.co.uk (C.H.); jon.barber@cefas.co.uk (J.B.); dave.sheahan@cefas.co.uk (D.S.); philippe.bersuder@cefas.co.uk (P.B.); adil.bakir@cefas.co.uk (A.B.)
2    Independent Researcher, Essex CM0 8QD, UK; chris.vivian2@btinternet.com
*    Correspondence: claire.mason@cefas.co.uk (C.M.); jemma.lonsdale@cefas.co.uk (J.-A.L.)

**Abstract:** Action Levels (ALs) are thresholds which are used to determine whether dredged material is suitable for disposal at sea by providing a proxy risk assessment for potential impacts to biological features such as fish and benthos. This project tested proposed scenarios for changes to the UK Action Levels to determine the likely implications for navigational dredge licensing in England and Wales. Approximately 3000 sample data records from 2009 to 2018 were collated with varying numbers of concentrations for contaminant parameters including trace metals, organotins, polycyclic aromatic hydrocarbons (PAHs), polychlorinated biphenyls (PCBs), organochlorine pesticides (OCPs) and polybrominated diphenyl ethers (PBDEs). Initially, these data were assessed using current ALs to determine the percentages of the samples with levels below AL1 (generally acceptable for disposal), between AL1 and AL2 and those showing levels above AL2 (generally unacceptable for disposal). These results were then used to compare with the results of the proposed new AL scenarios for each contaminant type derived from literature reviews and historic data. The results indicate that there are changes to the ALs which could be made such as updating the current ALs with the revised ALs, as well as the introduction of ALs where there are currently none set. The benefits of changing the ALs include reducing contaminant disposal to the marine environment and increased transparency in decision making. Any proposed scenarios will need to be phased in carefully in full liaison with stakeholders.

**Keywords:** sediment; contaminant; dredged material; action levels; disposal at sea

## 1. Introduction

As a seafaring island, the UK relies heavily on the maritime industry for the import and export of goods as well as for tourist and recreational activities to support the economy [1]. In order to keep the ports, harbours and marinas accessible for the aforementioned activities, regular clearance of the approach channels and berths are required through dredging [2]. Dredging can be for maintenance (regular dredging usually carried out annually) or for capital (the area has not been disturbed for at least ten years) [3] but both require a Marine Licence from the regulator (under the Marine and Coastal Access Act 2009 or the Planning Act 2008 depending on the particulars of the development).

Such dredging campaigns require an assessment of their potential impacts on the marine environment including chemical contamination [4]. In line with international obligations and guidelines, the UK assesses chemical contamination in sediments by comparison with Action Levels (ALs) [4,5]. ALs are thresholds which are used to determine whether dredged material is suitable for disposal at sea by providing a proxy risk assessment for potential impacts to biological features such as fish and benthos. There are two thresholds: Action Level 1 (AL1) is the lower threshold and Action Level 2 (AL2) is the upper threshold. Sediments with contaminant concentrations lower than AL1 are generally considered

acceptable for disposal at sea, pending other considerations such as physical suitability for the disposal site and potential beneficial uses. Sediments with contaminant concentrations above AL2 are generally considered unacceptable for uncontrolled disposal at sea without special handing and containment [6]. Sediments with contaminant concentrations between AL1 and AL2 are evaluated using a weight of evidence approach. The current UK ALs were proposed by Cefas and implemented in 1995 for England and Wales [6]. The current ALs for naturally occurring elements were derived on the basis of expert judgement applied to frequency distributions of dredged material analyses obtained over several years. For other contaminants where there were insufficient data for frequency distributions, they had to be estimated from available information in the literature and taking into account data that were available to Cefas.

In 2003, Cefas proposed 'Revised Action Levels' following a two-year project [7] considering the concentrations of contaminants in dredged material and the range of the toxicological effects of these contaminants. They were derived in line with international guidance [5,8] and the current scientific research at that time. The Revised ALs recommended modest changes to the current AL1, and all current AL2 levels were proposed to be reduced, potentially barring some sediments from disposal at sea which had previously been permitted. Therefore, the licensing body at that time, the Department for the Environment, Fisheries and Rural Affairs (Defra) were concerned about the impact on industry and decided not to implement the revised action levels until a strategy of managing the disposal of contaminated dredged material was completed.

In 2005, Cefas derived background levels of naturally occurring elements in sediments from areas around the coast of England and Wales that were subject to dredging and disposal [9]. By comparing the current AL1 values with the background levels, the report concluded that the current AL1 values employed in the UK were appropriate for environmental protection purposes. However, it was also noted that given variability in background levels, it would be inappropriate to use single background levels across England and Wales to determine whether particular sediments were or were not close to background levels in a specific location.

In 2015, the Marine Management Organisation (MMO) commissioned a high-level review of current action level guidance used in the licensing of the disposal of dredged material to sea [6]. The report concluded that the existing guidance and action levels were not fit for purpose, i.e., in terms of the ability to avoid disposal of toxic sediments at sea and refusal of non-toxic sediment disposal; however, it did state that the current UK approach fulfils the legal obligations under the Oslo–Paris Convention for the Protection of the Marine Environment of the North-East Atlantic (OSPAR). It also acknowledged that the overall fitness for purpose of regulatory tools such as ALs are importantly defined by legislative requirements and policy objectives which may include a consideration of costs and proportionate regulation as well as environmental risk. The report recommended that the UK approach to ALs would benefit from a further, more detailed review of the ALs and guidance to establish whether they are fit for purpose given current policy and regulatory requirements.

This study aims to build on the recommendations of this study by [6]:

- Reviewing the current AL2 (and potentially the current AL1 in parallel) as the UK current AL2s are the least protective of the marine environment within the OSPAR nations, whilst considering the burden it would place on industry;
- Reviewing the current ALs using UK specific sediments;
- Providing an initial analysis of how updating ALs could impact the port industry.

## 2. Materials and Methods

### 2.1. Data Collection

To enable the assessment of impacts to future licence consents, a dataset of dredge material testing data from Marine Licence applications was collated. These data were from individual samples (i.e., not averaged data as reported to OSPAR and London Convention/

London Protocol Secretariats). It was considered important that individual sample data were obtained as this would allow an assessment of the possible partial impacts to licences (for example, exclusions to part of a dredge area).

Data were collated from licence applications from England, Scotland, Wales and Northern Ireland. No data were available from for the Isle of Man, Jersey or Guernsey, but the results and possible implications for consenting on these authorities are discussed in Section 3 based on reported dredge material returns data. This dataset includes samples where the results showed levels of contaminants above Action Level 2 and would have therefore been excluded for disposal at sea.

Data for England were obtained though the MMOs public register [10] and Cefas' own data records of Marine Licence applications.

Data from Wales were obtained from Cefas' data records. Cefas have traditionally been the analysing laboratory for all Welsh disposal applications.

Data for Scotland were received upon request from the Marine Scotland Licensing Operations Team (MS LOT). Scottish data were provided in a new common reporting format established by MS LOT in 2017. Therefore, only data from 2017 and 2018 were available for this review.

Data for Northern Ireland were obtained upon request from the Department of Agriculture, Environment and Rural Affairs (DAERA). The data provided were in the format of laboratory data reports, some of which were scanned PDF documents and did not have sample coordinate information. These data were excluded from the wider dataset.

### 2.2. Data Rationalisation and Validation

Coordinates of all samples were checked using a GIS application (QGIS version 3.2.1). All samples were plotted against a UK coastline file. Sample points inland or offshore were validated by checking the coordinates on the original data forms. Any sample results where coordinates could not be confirmed (i.e., coordinate corresponding to the dredge area identified) were removed from the data.

It is noted that different authorities have different requirements for methods of analysis and laboratory standards. In practice this means that there a range of analytical methods were employed throughout the data collected. Most notably for trace metals, many of the Scottish results were analysed using a total (hydrofluoric acid) digestion method. This method can liberate more metals bound to sediment particles than the partial digest method required in England and Wales. No corrections were made for the differing methods. The primary reason for not correcting the data was to duplicate, as far as possible, the decision-making processes employed by the various authorities. In discussions with Marine Scotland, it was apparent that they utilise the Action Levels on the data provided without correction for the analytical procedure and therefore it was considered most appropriate to consider the data as they are reported.

Data reported as below detection (e.g., <LOD) were rationalised to the detection limit (e.g., <0.2 was changed to 0.2). This was considered a reasonable approach given the project aims (i.e., identifying exceedances above the current and proposed Action Levels) with the assumption that any proposed action level would be set above the detection limit for any determinant.

### 2.3. Data Scenario Selection

Literature reviews were completed separately for each contaminant type and proposed data scenarios were identified based on these literature reviews (Table 1). The data were considered sample by sample and were independent of licensing applications.

**Table 1.** Data scenarios tested as part of the Action Level Review. Expanded details for each scenario are given in the Results section (Section 3). The columns indicate current ALs, revised ALs, and then details of other scenarios tested for each contaminant group.

| Contaminant Group | Current ALs | Revised ALs [6] | Scenario 1 | Scenario 2 | Scenario 3 |
|---|---|---|---|---|---|
| Trace metals | Y | Y | Regional AL1s | | |
| Organotins | Y | Y | Proposal 1: Revised AL1 Revised AL2/2 | Proposal 2: Revised AL1/2 Revised AL2/5 | |
| Polycyclic Aromatic Hydrocarbons (PAHs) | N | Y | Individual PAHs: Canadian ISQG AL1 PEL AL2 | Summed PAHs: LMW/HMW ERL AL1 ERM AL2 | Summed PAHs: Σ16PAHs |
| Polychlorinated biphenyls (PCBs) | Y | Y | Individual PCBs: German | Individual PCBs: EACs EAC/3 AL1 EAC AL2 | Summed PCBs: Σ25_PCBs, ΣICES7 German |
| Organochlorine pesticides (OCPs) | Y | N | Individual OCPs: German | | |
| Polybrominated diphenyl ethers (PBDEs) | N | N | Individual BDEs: Canadian FESG/3 AL1 FESG AL2 | | |

*2.4. Data Assessment Metrics*

Three metrics were developed to aid in the interpretation of the results and indicate relative levels of any changes observed (see below). These were statistically derived. All results tables include the total number of samples, so it was possible to calculate the actual number of samples affected.

2.4.1. Below Action Level 1

This was calculated by subtracting the percentage number of samples below current AL1 from the scenario percentage number of samples below AL1 (Table 2). Scenario AL1 (% number of samples) − current AL1 (% number of samples) = % number of samples affected. Minus numbers indicate that lower numbers of samples were below AL1 (more protective) and positive numbers indicate that higher numbers of samples were below AL1 (more permissive).

**Table 2.** Metrics used to interpret scenario comparisons with the current AL1.

| Metric Descriptor | Range |
|---|---|
| Lower number of samples fall below AL1—more protective | >−10% |
| Slightly lower number of samples fall below AL1 | >−5% to −10% |
| Neutral | <−5% to <5% |
| Slightly higher number of samples fall below AL1 | >5% to 10% |
| Higher number of samples fall below AL1—more permissive | >10% |

2.4.2. Above AL2

This was calculated by subtracting the percentage number of samples above the current AL2 from the scenario percentage number of samples above AL2 (Table 3). Scenario AL2 (% number of samples) − current AL2 (% number of samples) = % number of samples affected. Minus numbers indicate that lower numbers of samples were above AL2 (more permissive) and positive numbers indicate that higher numbers of samples were above AL2 (more protective).

**Table 3.** Metrics used to interpret scenario comparisons for the current AL2.

| Metric Descriptor | Range |
|---|---|
| Lower number of samples fall above AL2—more permissive | >−10% |
| Slightly lower number of samples fall above AL2 | >−5% to −10% |
| Neutral | <−5% to <5% |
| Slightly higher number of samples fall above AL2 | >5% to 10% |
| Higher number of samples fall above AL2—more protective | >10% |

2.4.3. Range of Values

As indicated in the 'High-level Review' [6] the range of values between the current AL1 and current AL2 should be looked at, and reduced, if possible, to reduce the number of samples falling between the current AL1 and current AL2 and then requiring expert judgement. The difference in the range of values was calculated by subtracting the current range from the scenario range being tested (Table 4).

**Table 4.** Metrics used to help interpret scenario comparisons—range of values.

| Metric Descriptor | Range |
|---|---|
| Range reduced compared with current ALs | − |
| Range unchanged | |
| Range increased compared with current ALs | + |

**3. Results**

The samples within scope were assessed against the current ALs to provide a basis of comparison with the proposed data scenarios being tested. Results for the revised actions levels [6] and then the other proposed scenarios (Table 1) are presented for each contaminant group. These represent examples to use as a guide to help with filling in gaps and updating knowledge since the revised action levels were produced and to help with an understanding of the different approaches. The same metrics were used for assessing the differences in scenarios to current ALs (Section 2.4).

*3.1. Current Action Levels*

In the case of trace metals, up to 3% of the sample results were above the current AL2 (Table 5). The proportion of the sample results between the current AL1 and AL2 varied between 32% and 75% whilst the proportion of samples below AL1 varied between 25% and 66%. Trace metal analysis routinely includes arsenic, a non-metal. Arsenic was included with trace metals in this review even though it is a non-metal.

For organotins, up to 2% of the sample results were above the current AL2 (Table 5). ~85% of sample results were below AL1 for tributyltin (TBT) reflecting the reduction in concentrations observed around the coast since the cessation of the use of antifouling paints on shipping, as highlighted in the reduction in TBT concentrations at disposal sites [11].

There are no current ALs for polycyclic aromatic hydrocarbons (PAHs), although there is an AL1 for Total Hydrocarbons (THC) [7]. However, PAH concentrations are currently assessed using the proposed revised AL1s [7] as a guide. Results from the revised action level scenario are presented in Section 3.4.1.

The polychlorinated biphenyls (PCB) current ALs are for the sum of the concentration of all 25 PCBs ($\Sigma 25\_PCBs$)), as well as an AL1 for the $\Sigma ICES7$ [12]. More than 60% of samples were below the current AL1, and for current AL2 $\Sigma 25\_PCBs$ ~1% were above the current AL2 (Table 5). This is low, especially considering that the PCB sample analysis is only requested from areas that potentially have contamination.

**Table 5.** Current Action Levels (ALs) results—Percentage of samples below AL1; between AL1 and AL2, and above AL2 and range of values between AL1 and AL2. No current ALs for Polycyclic Aromatic Hydrocarbons (PAHs) or Polybrominated diphenyl ethers (PBDEs).

| Contaminant Group—Units for Current AL1, Current AL2 and Range | Contaminant | Current AL1 | Current AL2 | Range (AL2–AL1) | Sample Number below AL1 (%) | Sample Number above AL1/below AL2 (%) | Sample Number below AL2 (%) | Total Number of Samples |
|---|---|---|---|---|---|---|---|---|
| Trace Metals [1] —ppm | Arsenic (As) | 20 | 100 | 80 | 54 | 45 | 1 | 2719 |
| | Cadmium (Cd) | 0.4 | 5 | 4.6 | 66 | 33 | 1 | 2722 |
| | Chromium (Cr) | 40 | 400 | 360 | 45 | 55 | <1 | 2729 |
| | Copper (Cu) | 40 | 400 | 360 | 57 | 42 | 1 | 2734 |
| | Mercury (Hg) | 0.3 | 3 | 2.7 | 65 | 34 | 1 | 2699 |
| | Nickel (Ni) | 20 | 200 | 180 | 25 | 75 | <1 | 2724 |
| | Lead (Pb) | 50 | 500 | 450 | 48 | 51 | 1 | 2731 |
| | Zinc (Zn) | 130 | 800 | 670 | 45 | 52 | 3 | 2733 |
| Organotins —ppm | Dibutyltin (DBT) | 0.1 | 1 | 0.9 | 97 | 3 | <1 | 2147 |
| | Tributyltin (TBT) | 0.1 | 1 | 0.9 | 85 | 13 | 2 | 2234 |
| Polychlorinated biphenyls (PCBs) —ppb | Σ25_PCBs [2] | 20 | 200 | 180 | 63 | 36 | 1 | 915 |
| | ΣICES7 [3] | 10 | No AL2 | na [4] | 66 | 34 | na [4] | 1005 |
| Organo-chlorine pesticides (OCPs) —ppb | Dieldren | 5 | No AL2 | na [4] | 91 | 9 | na [4] | 186 |
| | Dichlorodipheny-ltrichloroethane (*p,p'*-DDT) | 1 | No AL2 | na [4] | 63 | 37 | na [4] | 181 |

[1] As is a non-metal included alongside trace metals. [2] Σ25_PCBs is the sum of measured PCBs: PCB101, PCB105, PCB110, PCB118, PCB128, PCB138, PCB141, PCB149, PCB151, PCB153, PCB156, PCB158, PCB170, PCB18, PCB180, PCB183, PCB187, PCB194, PCB28, PCB31, PCB44, PCB47, PCB49, PCB52 and PCB66. [3] ΣICES7 is the sum of ICES7 PCBs (ICES, 1990): PCB28, PCB52, PCB101, PCB118, PCB138 and PCB180. [4] na—not applicable as no AL2.

For organoclorine pesticides (OCPs), current AL1s are only defined for two pesticides, dieldrin and dichlorodiphenyltrichloroethane (DDT)). >90% of the sample results were below the current AL1 for dieldrin, with >60% for DDT (Table 5). It is assumed that this is related to the ban in the use of these pesticides.

There are no current ALs for polybrominated diphenyl ethers (PBDEs). However, PBDEs sample analysis is only requested from areas requiring capital dredge or in a limited number of applications that are known to potentially have contamination.

### 3.2. Trace Metals

#### 3.2.1. Revised Action Levels

The revised ALs [7] for metals (Table 6) show that there was a minimal change in the percentage of the sample results from the dataset above AL2. However, for the revised AL1 for the two metals chromium and nickel where the levels increased, there was a greater number of samples that fell below it (487 and 772 samples, respectively); therefore, for these two metals, the revised levels are more permissive (orange on Table 6). Conversely, for copper and mercury, there is a reduction in the concentration in the revised AL1 and fewer sample results fell below it, and so this is considered to be more protective (green on Table 6).

**Table 6.** Trace metals—Difference (revised − current) in percentage of samples below AL1; above AL2 for revised ALs and range of values between AL1 and AL2. Please note values (green) are more protective and values (orange) are more permissive (colour codes defined in Tables 2 to 4).

| Trace Metals [1] | Revised AL1 [7] (ppm) | Difference (ppm) | Difference in Sample Number below AL1 (%) | Revised AL2 [7] (ppm) | Difference (ppm) | Difference in Sample Number below AL2 (%) | Revised Range (ppm) | Difference in Range |
|---|---|---|---|---|---|---|---|---|
| Arsenic (As) | 20 | 0 | 0 | 70 | −30 | 1.5 | 80 | −30 |
| Cadmium (Cd) | 0.4 | 0 | 0 | 4 | −1 | 0.4 | 3.6 | −1 |
| Chromium (Cr) | 50 | 10 | 17.8 | 370 | −30 | 0 | 320 | −40 |
| Copper (Cu) | 30 | −10 | −15.2 | 300 | −100 | 0.5 | 360 | −90 |
| Mercury (Hg) | 0.25 | −0.05 | −7.6 | 1.5 | −1.5 | 2.4 | 1.25 | −1.45 |
| Nickel (Ni) | 30 | 10 | 28.3 | 150 | −50 | 0.1 | 120 | −60 |
| Lead (Pb) | 50 | 0 | 0 | 400 | −100 | 1 | 350 | −100 |
| Zinc (Zn) | 130 | 0 | 0 | 600 | −200 | 2.8 | 470 | −200 |

[1] Current ALs and the total number of samples are included in Table 5.

The revised AL2 for trace metals is lower than the current AL2 (Table 6), and thus proposes an increase in protection to environment, with relatively little change in the number of samples affected. The percentage of samples affected ranged from zero to 2.82% with zinc having the highest number of samples above the revised AL2 [7]. The range of concentrations for the revised ALs was smaller than for the current ALs for all trace metals, therefore reducing reliance on expert judgement (blue on Table 6).

These differences are not concentrated in one geographical area. Samples above AL2 are indicated in red. When comparing the proportion of sample results above AL2, between the revised and current ALs there was an increase in red markers for arsenic in the South West (Figure 1), for copper (Cu) in the South West and on the South coast (Figure 2), for mercury on the northeast coast (Figure 3), and for zinc in the North West, South West and the South Wales coast (Figure 4).

3.2.2. Regional Trace Metal Action Level 1s

Regional background levels for metals have been defined in a previous project [9]. Background levels from ports were proposed. The current AL1 for the trace metal determinants is two to three times the mean equivalent background value, so to derive regional action level 1 (regional AL1), three values were assessed in this scenario (regional background, regional background X2 and regional background X3).

The dataset and the background dataset were assigned to regions (Figure 5) based on river basins. The average background for each region was calculated for this demonstration, and these were used to compare with the current ALs. No regional background concentrations were developed for mercury. Figure 2 shows background, background X2, and background X3 for each metal with the current and revised AL1 and AL2 to put these into context (Figure 6). The effect of utilising regional background as a potential AL1 was assessed by comparing the percentage number of samples below the current ALs with the percentage number of samples below regional background, regional background X2 and regional background X3 (Table 7). Note that for all cases, regional background is most protective and regional background X3 is the most permissive.

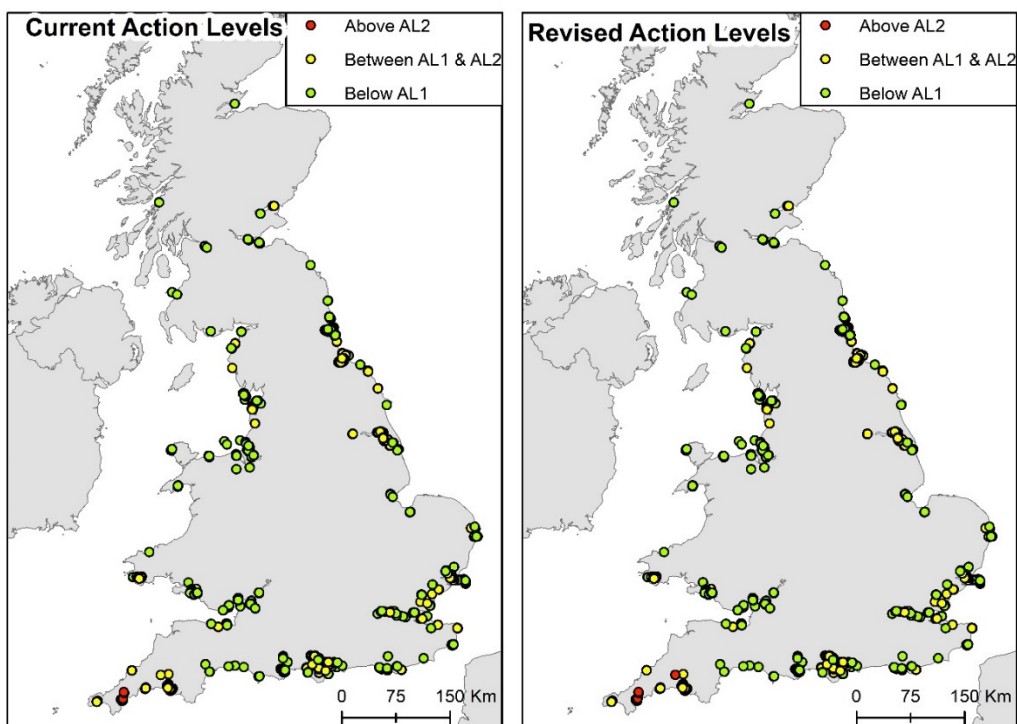

**Figure 1.** Maps showing the differences between the current ALs and revised ALs for arsenic (As). Samples above AL2 are indicated in red. Figure adapted from [13].

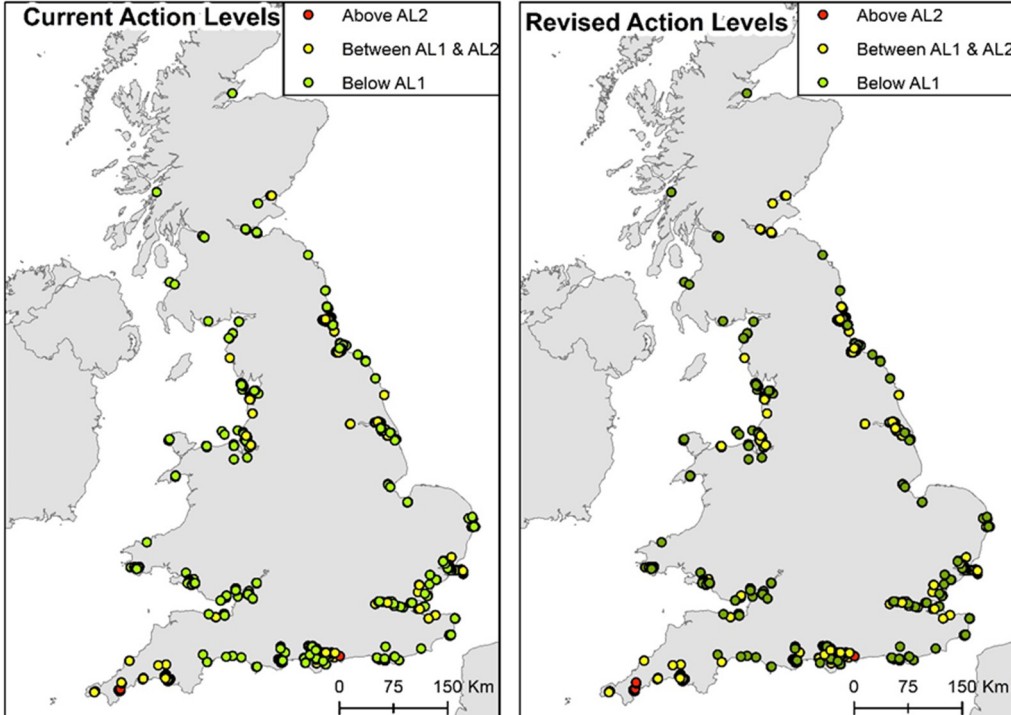

**Figure 2.** Maps showing the differences between the current ALs and revised ALs for copper (Cu). Samples above AL2 are indicated in red. Figure adapted from [13].

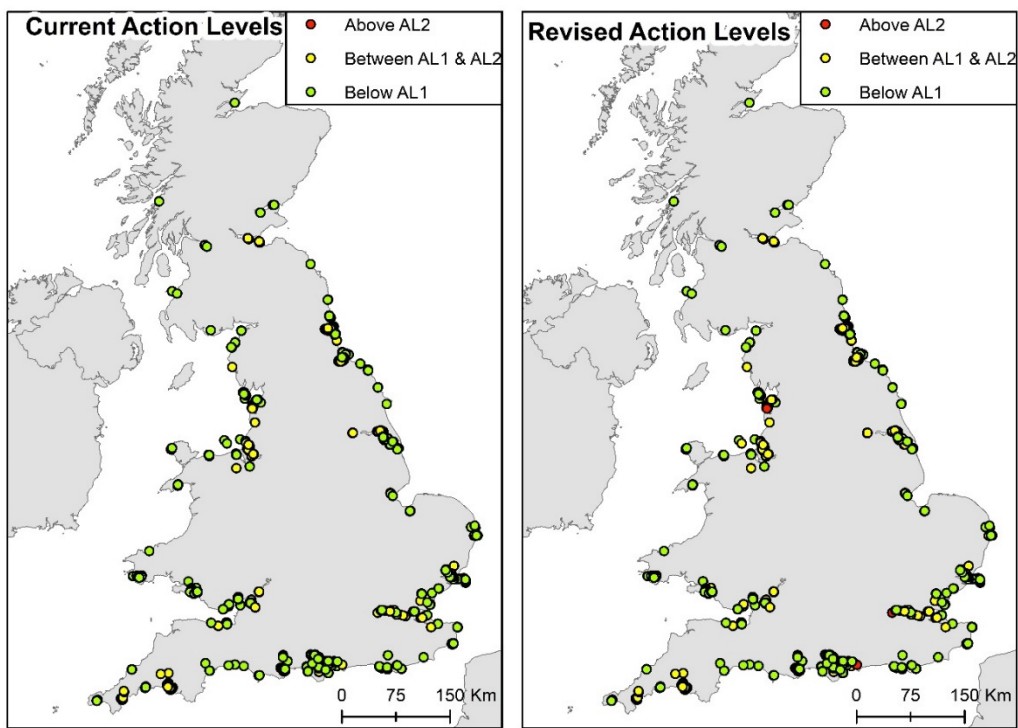

**Figure 3.** Maps showing the differences between the current ALs and revised ALs for mercury (Hg). Samples above AL2 are indicated in red. Figure adapted from [13].

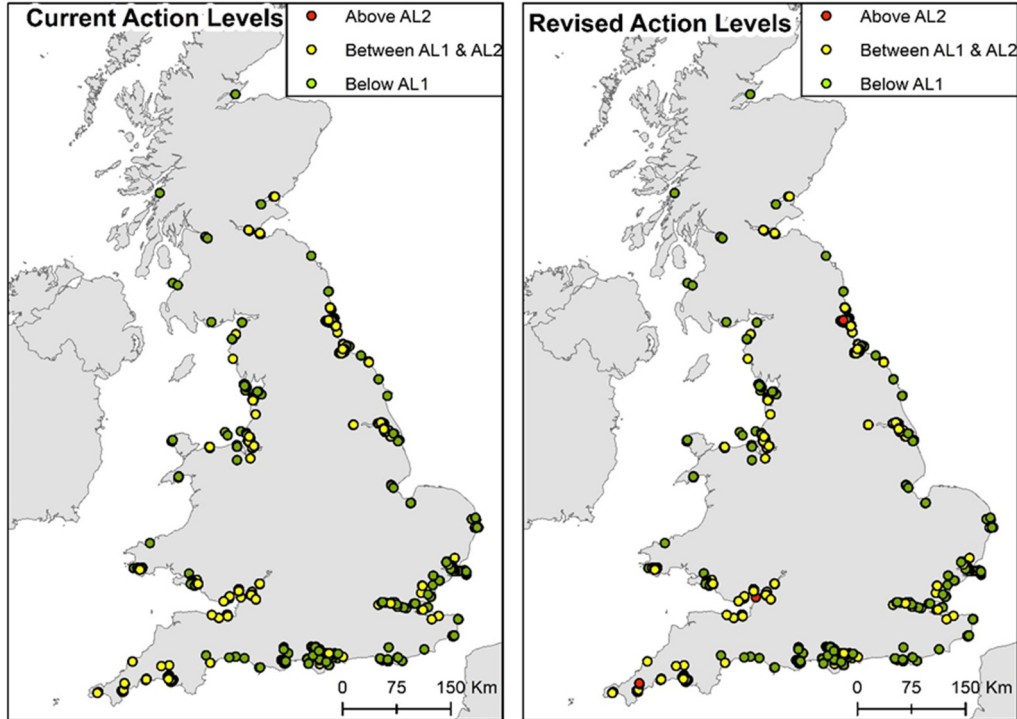

**Figure 4.** Maps showing the differences between the current ALs and revised ALs for zinc (Zn). Samples above AL2 are indicated in red. Figure adapted from [13].

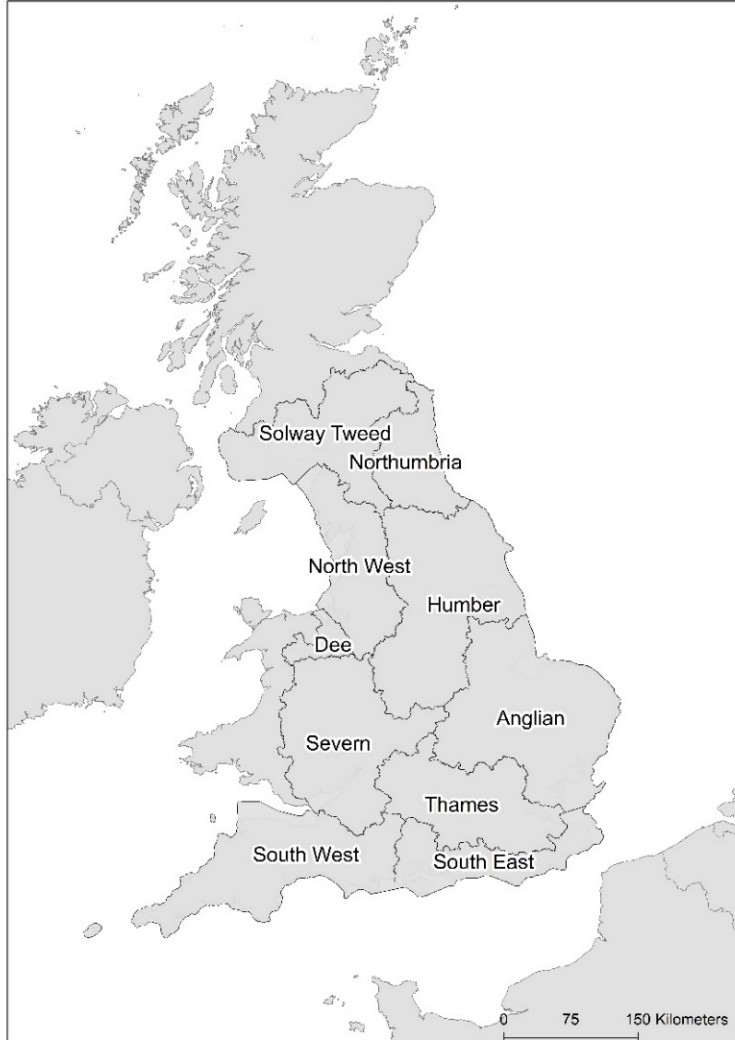

**Figure 5.** Map showing the regions used for testing regional metals action levels [9]. Figure adapted from [13].

The results showed that for Cd in all regions, the regional baseline was more permissive and would need revising, with some of the proposed background concentrations being close or more than the current AL2 (Figure 6). It is noted that the cadmium values need to be viewed as tentative since there no core data were available [9]. For the other metals, the values were more comparable with the current ALs (Figure 6) and it is clear introducing regional AL1 may be a good method to account for regional concentration differences (Table 7).

*3.3. Organotins*

3.3.1. Revised Action Levels

Applying revised ALs [7] for organotins (Table 8) show that there is likely to be a very slight increase in the number of samples above the revised AL2 for tributyltin (TBT) (44 samples out of the 2234 total number of samples or a maximum of a 2.1% increase) when compared to the number above the current AL2 in the dataset. The range of values for revised ALs was less than for the current ALs. Some samples in the South West and Humber were above the revised AL2 when compared with current ALs (Figure 7).

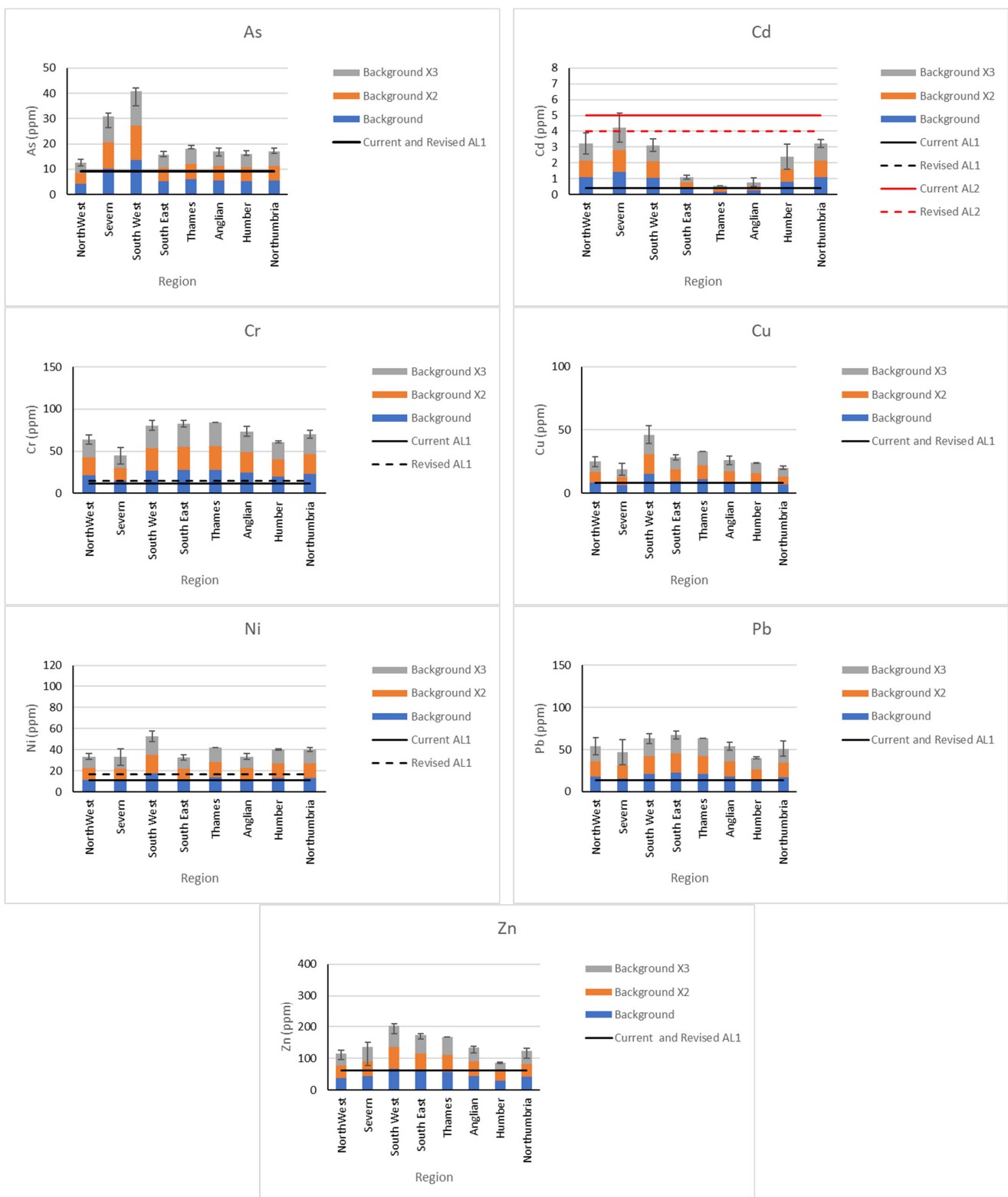

**Figure 6.** Trace metals—background, background X2, background X3 for each region with the current and revised AL1s (if different) indicated (current and revised AL2 shown only for cadmium (Cd). Note that if the current AL1 and revised AL1 are the same, only the AL1 shown. Error bars (95% confidence limits calculated by standard deviation X number of samples X 0.05) are indicated as a line on each stacked bar and apply to the whole bar. No regional background concentrations were developed for mercury (Hg). Regions are in geographic order starting from the Northwest coast ending in Northumbria (Figure 5). Figure adapted from [13].

**Table 7.** Trace metals—Regional background, and differences in numbers of samples (%) below AL1 (current AL1 − regional AL1) when compared to the current AL1 for 1/ regional background; 2/ regional background X2; and 3/ regional background X3. Regions are in geographic order starting from the North West coast ending in Northumbria (Figure 5). The number of samples (*n*) that are averaged for each metal regional background are indicated. Please note values (green) are more protective and values (orange) are more permissive (colour codes defined in Table 2).

| | | Arsenic (As) | Cadmium (Cd) [1] | Chromium (Cr) | Copper (Cu) | Nickel (Ni) | Lead (Pb) | Zinc (Zn) |
|---|---|---|---|---|---|---|---|---|
| | Current AL1 (ppm) | 20 | 0.40 | 40 | 40 | 20 | 50 | 130 |
| North West | Regional background (*n* = 6 except Cd where *n* = 4) (ppm) | 4 | 1 | 21 | 8 | 11 | 18 | 39 |
| | Number of samples below current AL1 (%) | 61 | 56 | 46 | 57 | 33 | 47 | 43 |
| | 1/ regional background (difference (%)) | −56 | 35 | −22 | −37 | −14 | −21 | −26 |
| | 2/ regional background X2 (difference (%)) | −41 | 42 | 1 | −24 | 4 | −8 | −13 |
| | 3/ regional background X3 (difference (%)) | −25 | 44 | 25 | −11 | 33 | 1 | −3 |
| | Total number of samples | 228 | 228 | 230 | 231 | 229 | 230 | 230 |
| Severn | Regional background (*n* = 3) (ppm) | 10 | 1 | 15 | 6 | 11 | 16 | 45 |
| | Number of samples below current AL1 (%) | 88 | 72 | 17 | 85 | 6 | 7 | 4 |
| | 1/ regional background (difference (%)) | −86 | 28 | −15 | −84 | −5 | −7 | −4 |
| | 2/ regional background X2 (difference (%)) | 4 | 28 | −10 | −81 | −2 | −4 | −1 |
| | 3/ regional background X3 (difference (%)) | 12 | 28 | 7 | −81 | 21 | −3 | 0 |
| | Total number of samples | 118 | 118 | 118 | 118 | 118 | 118 | 118 |
| South West | Regional background (*n* = 7) (ppm) | 14 | 1 | 27 | 15 | 18 | 21 | 68 |
| | Number of samples below current AL1 (%) | 39 | 57 | 58 | 38 | 31 | 43 | 38 |
| | 1/ regional background (difference (%)) | −18 | 35 | −28 | −22 | −4 | −26 | −20 |
| | 2/ regional background X2 (difference (%)) | 12 | 42 | 26 | −7 | 33 | −7 | 3 |
| | 3/ regional background X3 (difference (%)) | 24 | 43 | 40 | 4 | 64 | 5 | 22 |
| | Total number of samples | 336 | 335 | 338 | 338 | 336 | 338 | 338 |
| South East | Regional background (*n* = 8) (ppm) | 5 | 0.37 | 28 | 9 | 11 | 23 | 58 |
| | Number of samples below current AL1 (%) | 58 | 91 | 54 | 63 | 39 | 77 | 72 |
| | 1/ regional background (difference (%)) | −56 | −0 | −26 | −59 | −30 | −52 | −55 |
| | 2/ regional background X2 (difference (%)) | −50 | 7 | 29 | −51 | 11 | −6 | −9 |
| | 3/ regional background X3 (difference (%)) | −28 | 7 | 45 | −34 | 54 | 9 | 16 |
| | Total number of samples | 713 | 714 | 714 | 716 | 714 | 715 | 716 |
| Thames | Regional background (*n* = 1) (ppm) | 6 | 0.19 | 28 | 11 | 14 | 21 | 56 |
| | Number of samples below current AL1 (%) | 75 | 64 | 55 | 66 | 48 | 56 | 63 |
| | 1/ regional background (difference (%)) | −74 | −18 | −12 | −39 | −16 | −25 | −36 |
| | 2/ regional background X2 (difference (%)) | −37 | 0 | 18 | −19 | 25 | −7 | −4 |
| | 3/ regional background X3 (difference (%)) | −4 | 16 | 37 | −6 | 34 | 7 | 13 |
| | Total number of samples | 101 | 102 | 102 | 102 | 102 | 102 | 102 |
| Anglian | Regional background (*n* = 5) (ppm) | 6 | 0.26 | 25 | 9 | 11 | 18 | 45 |
| | Number of samples below current AL1 (%) | 55 | 89 | 33 | 72 | 19 | 83 | 72 |
| | 1/ regional background (difference (%)) | −43 | −14. | −16 | −59 | −8 | −59 | −58 |
| | 2/ regional background X2 (difference (%)) | −29 | 3 | 21 | −46 | 4 | −37 | −36 |
| | 3/ regional background X3 (difference (%)) | −14 | 8. | 58 | −15 | 42 | 6 | 2 |
| | Total number of samples | 198 | 198 | 199 | 200 | 198 | 200 | 200 |
| Humber | Regional background (*n* = 3) (ppm) | 5 | 0.80 | 20 | 8 | 13 | 13 | 29 |
| | Number of samples below current AL1 (%) | 32 | 76 | 33 | 63 | 16 | 30 | 100 |
| | 1/ regional background (difference (%)) | −29 | 12 | −17 | −55 | −7 | −23 | −97 |
| | 2/ regional background X2 (difference (%)) | −21 | 18 | 2 | −43 | 9 | −13 | −87 |
| | 3/ regional background X3 (difference (%)) | −11 | 22 | 22 | −32 | 45 | −4 | −79 |
| | Total number of samples | 240 | 240 | 240 | 240 | 240 | 240 | 240 |

**Table 7.** *Cont.*

| | | Arsenic (As) | Cadmium (Cd) [1] | Chromium (Cr) | Copper (Cu) | Nickel (Ni) | Lead (Pb) | Zinc (Zn) |
|---|---|---|---|---|---|---|---|---|
| Northumbria | Regional background (*n* = 3) (ppm) | 6 | 1.08 | 23 | 7 | 13 | 17 | 41 |
| | Number of samples below current AL1 (%) | 37 | 21 | 33 | 30 | 9 | 7 | 10 |
| | 1/ regional background (difference (%)) | −35 | 28 | −25 | −27 | −4 | −5 | −9 |
| | 2/ regional background X2 (difference (%)) | −30 | 56 | 14 | −25 | 10 | −3 | −4 |
| | 3/ regional background X3 (difference (%)) | −9 | 70 | 45 | −21 | 71 | 0 | −0 |
| | Total number of samples | 493 | 495 | 496 | 497 | 495 | 496 | 497 |

[1] Cadmium (Cd) regional backgrounds should be viewed as tentative as no core data were available [9].

**Table 8.** Organotins (DBT, dibutyltin; TBT, tributyltin)—Difference (revised − current) in the percentage of samples below AL1; above AL2 for revised ALs and the range of values between AL1 and AL2. Please note values (green) are more protective and values (orange) are more permissive (colour codes defined in Tables 2 to 4).

| Organotins | Revised AL1 [7] (ppm) | Difference (ppm) | Difference in Sample Number below AL1 (%) | Revised AL2 [7] (ppm) | Difference (ppm) | Difference in Sample Number below AL2 (%) | Range (Revised) | Difference in Range |
|---|---|---|---|---|---|---|---|---|
| Dibutyltin (DBT) [1] | 0.1 | 0 | 0 | 0.5 | −0.5 | 0.2 | 0.4 | −0.5 |
| Tributyltin (TBT) [1] | 0.1 | 0 | 0 | 0.5 | −0.5 | 2 | 0.4 | −0.5 |

[1] Current ALs, and Total number of samples are included in Table 5.

### 3.3.2. Proposed Scenarios

The two scenarios for organotins look to reduce both the ALs for dibutyltin (DBT) and tributyltin (TBT). As both ALs were reduced from the current ALs, there was a greater number of samples that exceeded the current AL2, but as the range between AL1 and AL2 was reduced, there was also a greater number of samples that fell below the current AL1 (Table 9). There was no difference between the percentage of the samples that fell below AL1 for Proposal 1 (Table 9).

### 3.4. Polycyclic Aromatic Hydrocarbons (PAHs)

### 3.4.1. Revised Action Levels

There are currently no ALs for PAHs and so the revised ALs [7] for PAHs (Table 10) show the percentage number of samples that would fall below AL1 if implemented. It should be noted that while the revised AL1s [7] for PAHs were for 17 individual PAHs, here we have extended this to include the 22 individual/cluster PAHs that are routinely analysed for marine licensing. The results show that there was a variation in the number of samples below the revised ALs, ranging from 17% (*n* = 226) for dibenz (a,h) anthracene to 89% (*n* = 1182) for acenaphthene.

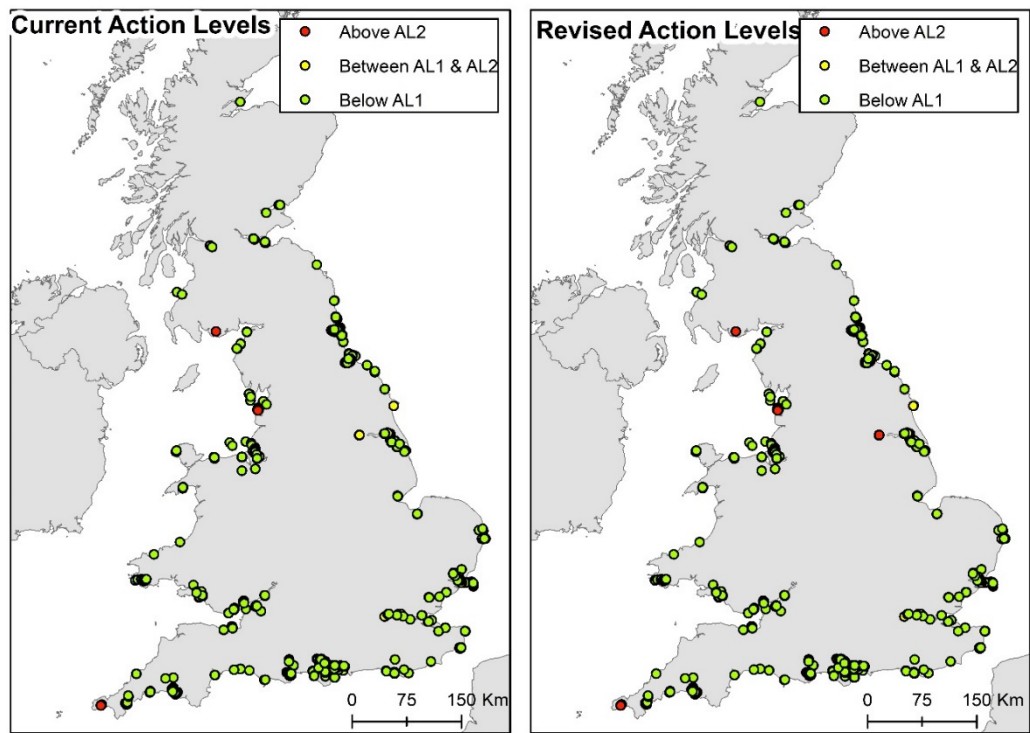

**Figure 7.** Maps showing the differences between the current ALs and revised ALs for organotins (DBT and TBT). Samples above AL2 are indicated in red. Figure adapted from [13].

**Table 9.** Organotins (DBT, dibutyltin; TBT, tributyltin)—Proposals 1 and 2 Difference (proposal − current) in the percentage of samples below AL1, above AL2 for revised ALs and the range of values between AL1 and AL2. Please note values (green) are more protective and values (orange) are more permissive (colour codes defined in Tables 2 to 4).

| Proposal 1: Revised AL1, Revised AL2/2 Organotins (ppm) | Proposal 1 AL1 (ppm) | Difference (ppm) | Difference in Sample Number below AL1 (%) | Proposal 1 AL2 (ppm) | Difference (ppm) | Difference in Sample Number below AL2 (%) | Proposal 1 Range (ppm) | Difference in Range |
|---|---|---|---|---|---|---|---|---|
| Dibutyltin (DBT) [1] | 0.1 | 0 | na | 0.25 | −0.75 | 1 | 0.15 | −0.75 |
| Tributyltin (TBT) [1] | 0.1 | 0 | na | 0.25 | −0.75 | 5 | 0.15 | −0.75 |

| Proposal 2: Revised AL1/2, Revised AL2/5 Organotins (ppm) | Proposal 2 AL1 (ppm) | Difference (ppm) | Difference in Sample Number below AL1 (%) | Proposal 2 AL2 (ppm) | Difference (ppm) | Difference in Sample Number below AL2 (%) | Proposal 2 Range (ppm) | Difference in Range |
|---|---|---|---|---|---|---|---|---|
| Dibutyltin (DBT) [1] | 0.05 | −0.05 | −5 | 0.1 | −0.9 | 3 | 0.05 | −0.85 |
| Tributyltin (TBT) [1] | 0.05 | −0.05 | −10 | 0.1 | −0.9 | 13 | 0.05 | −0.85 |

[1] Current ALs, and Total number of samples are included in Table 5.

**Table 10.** Polycyclic Aromatic Hydrocarbons (PAHs)—Revised—Individual PAHs and Total Hydrocarbons (THC). There are no current AL1s, AL2s or revised AL2s. The percentage of samples below AL1 and the percentage of samples below AL1 with the Northumbria and Humber samples excluded (NE excluded) are presented.

| Polycyclic Aromatic Hydrocarbons (PAHs) | | Revised AL1 [7] (ppb) | Number of Samples below AL1 (%) | Number of Samples below AL1 (%)—NE Excluded) | Total Number of Samples | Total Number of Samples —NE Excluded |
|---|---|---|---|---|---|---|
| **Abbreviation** | **Full Name** | | | | | |
| Acenapth | Acenaphthene | 100 | 89 | 95 | 1861 | 1328 |
| Acenapthylene | Acenaphthylene | 100 | 75 | 88 | 1870 | 1335 |
| Anthracn | Anthracene | 100 | 51 | 66 | 1869 | 1334 |
| BAA | Benz(a)anthracene | 100 | 30 | 38 | 1865 | 1330 |
| BAP | Benzo(a)pyrene | 100 | 27 | 34 | 1872 | 1337 |
| BBF | Benzo(b)fluoranthene | 100 | 22 | 27 | 1873 | 1338 |
| BEP | Benzo(e)pyrene | 100 | 30 | 38 | 1850 | 1316 |
| Benzghip | Benzo(g,h,i)perylene | 100 | 30 | 37 | 1872 | 1337 |
| BKF | Benzo(k)fluoranthene | 100 | 37 | 46 | 1865 | 1330 |
| C1N | Methyl Naphthalenes | 100 | 37 | 52 | 1807 | 1273 |
| C1PHEN | Methyl Phenanthrenes/Anthracenes | 100 | 22 | 31 | 1806 | 1273 |
| C2N | Dimethyl Naphthalenes | 100 | 27 | 38 | 1804 | 1271 |
| C3N | Trimethyl Naphthalenes | 100 | 20 | 27 | 1802 | 1270 |
| Chrysene | Chrysene | 100 | 34 | 43 | 1860 | 1328 |
| Debenzah | Dibenz[a,h]anthracene | 100 | 17 | 21 | 1860 | 1328 |
| Flurant | Fluoranthene | 100 | 19 | 23 | 1865 | 1333 |
| Fluorene | Fluorene | 100 | 59 | 76 | 1860 | 1329 |
| Indypr | Indeno(1,2,3-cd)pyrene | 100 | 27 | 33 | 1860 | 1329 |
| Napth | Naphthalene | 100 | 48 | 64 | 1862 | 1331 |
| Perylene | Perylene | 100 | 36 | 44 | 1842 | 1311 |
| Phenant | Phenanthrene | 100 | 30 | 40 | 1861 | 1330 |
| Pyrene | Pyrene | 100 | 18 | 22 | 1860 | 1329 |
| THC | Total Hydrocarbons | 100 (ppm) | 15 | 80 | 2048 | 1429 |

When the samples from Northumbria and Humber (areas known to be heavily impacted by historical industrial activities) were excluded from the data set for PAHs [14], then a greater percentage of sample results fell below AL1 than when all sample results were considered (Table 10).

Total Hydrocarbons (THC) are currently used informally to assess whether dredged material can be disposed of to sea by the regulators and their scientific advisors. However, the use of THC is limited as it provides no indication of toxicity, there is large inter-laboratory method variability and it appears to be conservative given that most sediment fails this threshold leading to additional evidence and assessments being required.

### 3.4.2. Proposed Scenarios

There are currently no ALs for PAHs; therefore, the percentage number of sample results below the proposed AL1 and above the proposed AL2 in the scenarios are presented for individual PAHs using the Canadian thresholds Interim Sediment Quality Guidelines [14] (ISQG) for AL1 andpermissible exposure levels (PELs) for AL2 (Table 11); and for summed PAHs for both LMW/HMW [14] ERLs for AL1 and ERMs for AL2 (Table 12) and for both ALs for $\sum$16PAH [15] (Table 13) were completed. These scenarios were repeated

with the samples in Northumbria and Humber excluded as these regions have elevated levels of hydrocarbons. As can be seen, the results for the percentage of samples that exceeded AL2 was variable depending on the individual PAH. The results show that when data from Northumbria and the Humber were excluded, the percentage of samples exceeding AL2 was reduced.

**Table 11.** Polycyclic Aromatic Hydrocarbons (PAHs)—Individual PAHs—Canadian ISQG/PEL [16]. Percentage of samples below AL1(ISQG) and above AL2 (PEL) and the respective percentage of samples with Northumbria and Humber samples excluded (NE excluded) are presented.

| Polycyclic Aromatic Hydrocarbons (PAHs) —Abbreviation [1] | ISQG [13] AL1 (ppb) | Number of Samples below AL1 (%) | Number of Samples below AL1 (%)—NE Excluded | PEL [13] AL2 (ppb) | Number of Samples above AL2 (%) | Number of Samples below AL2 (%)—NE Excluded |
|---|---|---|---|---|---|---|
| Acenapth | 7 | 31 | 38 | 89 | 12 | 6 |
| Acenapthylene | 6 | 21 | 27 | 128 | 21 | 9 |
| Anthracn | 47 | 38 | 49 | 245 | 25 | 12 |
| BAA | 75 | 24 | 30 | 693 | 23 | 12 |
| BAP | 89 | 24 | 30 | 763 | 21 | 14 |
| Chrysene | 108 | 35 | 44 | 846 | 12 | 8 |
| Debenzah | 6 | 14 | 18 | 135 | 23 | 12 |
| Flurant | 113 | 20 | 25 | 1494 | 19 | 11 |
| Fluorene | 21 | 31 | 42 | 144 | 31 | 14 |
| Napth | 35 | 31 | 42 | 391 | 27 | 6 |
| Phenant | 87 | 27 | 37 | 544 | 34 | 15 |
| Pyrene | 153 | 26 | 32 | 1398 | 19 | 11 |

[1] Full names and total numbers of samples are included in Table 10.

**Table 12.** Polycyclic Aromatic Hydrocarbons (PAHs)—Summed PAHs—LMW and HMW—ERL and ERMs [15]. Percentage of samples below AL1(ERL) and above AL2 (ERM) and the respective percentage of samples with Northumbria and Humber samples excluded (NE excluded) are presented.

| Polycyclic Aromatic Hydrocarbons (PAHs) | ERL [15] AL1 (ppb) | Number of Samples below AL1 (%) | Number of Samples below AL1 (%)—NE Excluded | ERM [15] AL2 (ppb) | Number of Samples below AL2 (%) | Number of Samples below AL2 (%)—NE Excluded | Total Number of Samples | Total Number of Samples—NE Excluded |
|---|---|---|---|---|---|---|---|---|
| LMW [1] | 552 | 40 | 54 | 3160 | 24 | 6 | 1874 | 1339 |
| HMW [2] | 1700 | 47 | 60 | 9600 | 8 | 6 | 1875 | 1340 |

[1] LMW is the Sum of Naphthalene, Acenaphthene, Fluorene, Anthracene, C1-naphtha-lenes, Acenaphthylene, Phenanthrene. [2] HMW is the Sum of Fluoranthene, Pyrene, Benz(a)anthracene, Chrysene, Benzo(a)pyrene, Dibenz(a,h)[a,b] anthracene.

**Table 13.** Polycyclic Aromatic Hydrocarbons (PAHs)—Summed PAHs—Sum of 16 Polycyclic Aromatic Hydrocarbons (Σ16PAH) [17]. Percentage of samples below AL1 and the percentage of samples with Northumbria and Humber samples excluded (NE excluded) are presented.

| Polycyclic Aromatic Hydrocarbons (PAHs) | ERL [15] AL1 (ppb) | Number of Samples below AL1 (%) | Number of Samples below AL1 (%)—NE Excluded | ERM [15] AL2 (ppb) | Number of Samples below AL2 (%) | Number of Samples below AL2 (%)—NE Excluded | Total Number of Samples | Total Number of Samples—NE Excluded |
|---|---|---|---|---|---|---|---|---|
| Σ16PAH [1] [17] | 2000 | 37 | 47 | 45,000 | 2 | 1 | 1876 | 1341 |

[1] Σ16PAH is the Sum of Acenaphthylene, Acenaphthene, Anthracene, Benz(a)[a]anthracene, Benzo(a)[a]pyrene, Benzo(b)[b]fluoranthene, Benzo(g,h,i)perylene, Benzo(k)fluoranthene, Chrysene, Dibenz(a,h)anthracene, Fluoranthene, Fluorene, Indeno(1,2,3-cd)]pyrene, Naphthalene, Phenanthrene, Pyrene.

For summed LMW and HMW PAHs, ERLs and ERMs were the more sensitive scenarios tested (Figure 8) when compared with ∑16PAH [17] (Figure 9) where very few samples overall were above AL2. Figure 8 also demonstrates the regional significance, especially for LMW PAHs, in Northumbria and the Humber (Table 12).

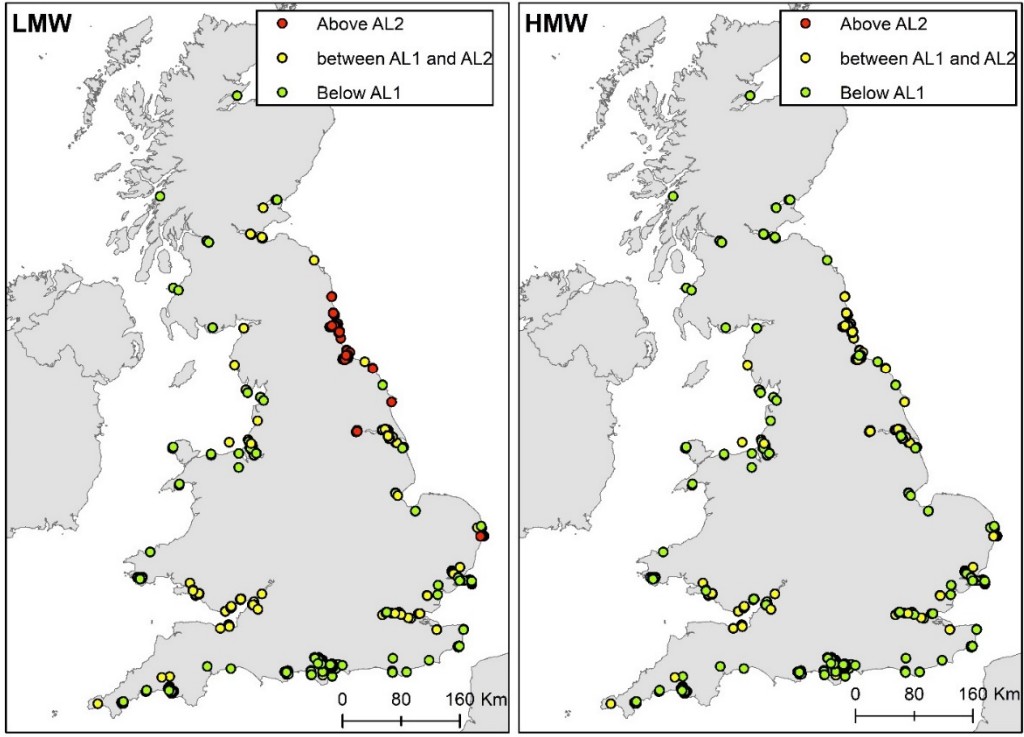

**Figure 8.** Maps showing ERLs (AL1) and ERMs (AL2) for LMW and HMW [15] summed PAHs (Table 12). Samples above AL2 are indicated in red. Figure adapted from [13].

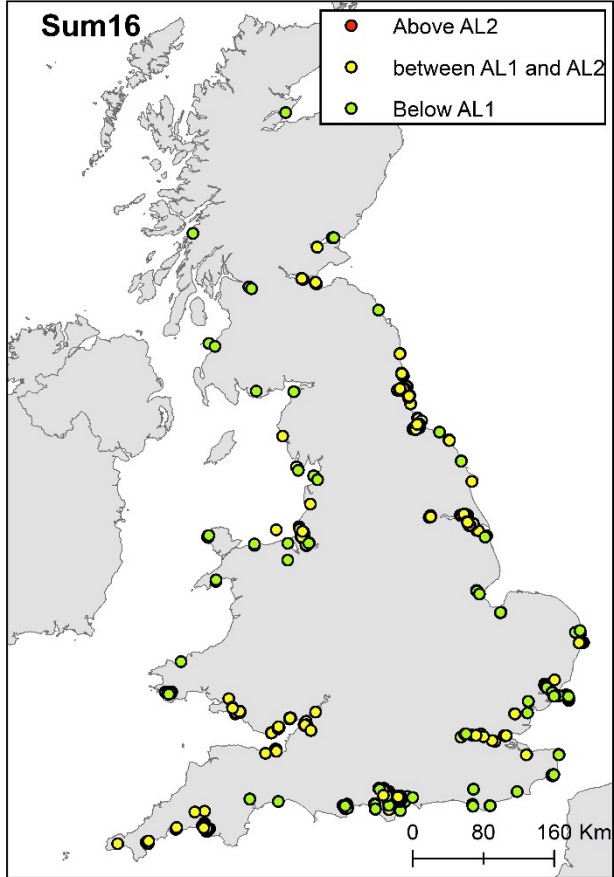

**Figure 9.** Maps showing summed 16PAHs [17] (Table 13). Samples above AL2 are indicated in red. Figure adapted from [13].

*3.5. Polychlorinated Biphenyls (PCBs)*

3.5.1. Revised Action Levels

When the revised ALs [7] for PCBs (Table 14) were applied to the data there was minimal difference in the number of sample results observed above AL2 for the Σ25_PCBs when compared with the number observed over the current AL2 (0.33% or three samples). The range of concentrations for the revised ALs was smaller than for the current ALs, reducing the need for expert judgement.

3.5.2. Proposed Scenarios

There are no current action levels for individual PCBs. Therefore, for individual PCBs, the percentage number of samples below AL1 and above AL2 are presented using German action levels, chosen from the OSPAR Overview of Contracting Parties' National Action Levels for Dredged Material [8] as these were the most protective (Table 15). An alternative scenario using the Environmental Assessment Criteria (EAC), as used for offshore assessments by OSPAR [18], was used for AL2, and AL1 was derived from German approach to the AL2 by dividing the EAC/3 (Table 16). The results show that the percentage of samples that fell below AL1 and exceeded AL2 was similar across congeners with PCB52 showing the highest proportion of samples exceeding AL2 (~11% which equates to 104 of the 985 samples analysed). The scenario based on EACs had a lower range as well as a greater proportion of samples with concentrations greater than AL2. Here, CB118 had the highest proportion of samples exceeding AL2.

**Table 14.** Polychlorinated biphenyls (PCBs)—Summed PCBs—Σ25_PCBs—Difference (revised − current) in the percentage of samples below AL1; above AL2 for revised ALs and range of values between AL1 and AL2. Please note values (green) are more protective and values (orange) are more permissive (colour codes defined in Tables 1 to 3). Difference (revised − current) in the percentage of samples below AL1; above AL2 for revised ALs and range of values between AL1 and AL2. Please note values (green) are more protective and values (orange) are more permissive (colour codes defined in Tables 2 to 4).

| Polychlorinated Biphenyls (PCBs) | Revised AL1 [7] (ppm) | Difference (ppb) | Difference in Sample Number above AL1 (%) | Revised AL2 [7] (ppb) | Difference (ppb) | Difference in Sample Number above AL2 (%) | Range (Revised) (ppb) | Difference in Range |
|---|---|---|---|---|---|---|---|---|
| Σ25_PCBs [1] | 20 | 0 | 0 | 180 | −20 | −20 | 160 | −20 |

[1] Current ALs, Σ25_PCBs definition and Total number of samples are included in Table 5.

**Table 15.** Polychlorinated biphenyls (PCBs)—Individual PCBs (OSPAR overview German — Percentage of samples below AL1, above AL2 and range.

| Polychlorinated Biphenyls (PCBs) | OSPAR Overview German [8] AL1 (ppb) | Number of Samples below AL1 (%) | OSPAR Overview German [8] AL2 (ppb) | Number of Samples above AL2 (%) | OSPAR Overview German [8] Range (ppb) | Total Number of Samples |
|---|---|---|---|---|---|---|
| PCB101 | 2 | 72 | 6 | 7 | 4 | 992 |
| PCB118 | 3 | 85 | 10 | 3 | 7 | 995 |
| PCB138 | 4 | 83 | 12 | 3 | 8 | 963 |
| PCB153 | 5 | 89 | 15 | 2 | 10 | 990 |
| PCB180 | 2 | 80 | 6 | 5 | 4 | 989 |
| PCB28 | 2 | 81 | 6 | 6 | 4 | 984 |
| PCB52 | 1 | 62 | 3 | 11 | 2 | 985 |

For summed scenarios for the Σ25_PCBs and ΣICES7 based on the German thresholds in the OSPAR overview [8], a higher proportion of samples fell below AL1 than the current ALs (Table 17). For ΣICES7, a proposed AL2 was tested, calculated by halving the revised Σ25_PCBs because AL1 for Σ25_PCBs is half of AL1 for ΣICES7. There were minimal differences in the numbers of samples below AL1 or above AL2 for these scenarios.

**Table 16.** Polychlorinated biphenyls (PCBs)—Individual PCBs (EACs) [18]—Percentage of samples below AL1; above AL2; and range.

| Polychlorinated Biphenyls (PCBs) | EAC/3 [17] AL1 (ppb) | Number of Samples below AL1 (%) | EAC [18] AL2 (ppb) | Number of Samples below AL2 (%) | EAC Range (ppb) | Total Number of Samples |
|---|---|---|---|---|---|---|
| PCB101 | 1 | 53 | 3 | 18 | 2 | 992 |
| PCB118 | 0.2 | 3 | 0.6 | 55 | 0.4 | 995 |
| PCB138 | 2.6 | 76 | 7.9 | 6 | 5.3 | 963 |
| PCB153 | 13 | 97 | 40 | 1 | 27 | 990 |
| PCB180 | 4 | 91 | 12 | 1 | 8 | 989 |
| PCB28 | 0.6 | 57 | 1.7 | 23 | 1.1 | 984 |
| PCB52 | 0.9 | 59 | 2.7 | 12 | 1.8 | 985 |

**Table 17.** Polychlorinated biphenyls (PCBs)—Summed PCBs—Σ25_PCBs and ΣICES7—Difference (proposed − current) in the percentage of samples below AL1; above AL2 for the proposed ALs and range of values between AL1 and AL2. Please note there is no current AL2 for ΣICES7, and values (green) are more protective and values (orange) are more permissive (colour codes defined in Tables 2 to 4).

| Polychlorinated Biphenyls (PCBs) | | Proposed AL1 (ppm) | Difference (ppb) | Difference in Sample Number below AL1 (%) | Proposed AL2 (ppb) | Difference (ppb) | Difference (Σ25_PCBs[1]) or or Number of Samples (ΣICES7[1]) above AL2 (%) | Range (ppb) | Difference in Range |
|---|---|---|---|---|---|---|---|---|---|
| Proposal 1 OSPAR Overview German [8] | Σ25_PCBs[1] | 40 | 20 | 17 | 120 | −80 | 2 | 80 | −100 |
| | ΣICES7[1] | 20 | 10 | 16 | 60 | na | 4 | 40 | na |
| Proposal 2: 1/2 Revised [7] as for Σ25_PCBs[1] | ΣICES7[1] AL2 | 10 | 0 | 0 | 90 | na | 2 | 80 | na |

[1] Current Als; Σ25_PCBs and ΣICES7 definitions; and Total number of samples are included in Table 5.

*3.6. Organochlorine Pesticides (OCPs)*

3.6.1. Revised Action Levels

There were no revised ALs for organochlorine pesticides (OCPs) [7].

3.6.2. Proposed Scenarios

There are only two current ALs for OCPs (dieldrin and dichlorodiphenyltrichloroethane (DDT)). Therefore, the percentage number of sample results from the scenarios below AL1 and above AL2 are presented (as for PCBs) using German values from OSPAR overview [8]. These were chosen as they were most protective for PCBs (Table 18).

**Table 18.** Organochlorine pesticides (OCPs)—Individual OCPs—OSPAR overview German—Percentage of samples below AL1; above AL2; and range. No data and no AL1 or AL2 are available for beta-Hexachlorocyclohexane (β-HCH) (BHCH); and no AL1 or AL2 are available for Dieldren (OSPAR overview German [8]).

| Organochlorine Pesticides (OCPs) | | OSPAR Overview German [8] AL1 (ppb) | Number of Samples below AL1 (%) | OSPAR Overview German [8] AL2 (ppb) | Number of Samples below AL2 (%) | Total Number of Samples |
|---|---|---|---|---|---|---|
| **Abbreviation** | **Name** | | | | | |
| AHCH | alpha-Hexachlorocyclohexane (α-HCH) | 0.4 | 88 | 1 | 3 | 167 |
| GHCH | gamma-Hexachlorocyclohexane (γ-HCH) (also known as lindane) | 0.2 | 0 | 0.6 | 5 | 167 |
| HCB | Hexachlorobenzene | 2 | 92 | 6 | 1 | 435 |
| DDE | Dichlorodiphenyldichloroethylene (*p,p'*-DDE) | 1 | 57 | 3 | 11 | 479 |
| DDT [1] | Dichlorodiphenyltrichloroethane (*p,p'*-DDT) [1] | 1 | 0 [1] | 3 | 18 | 181 |
| TDE | dichlorodiphenyldichloroethane (*p,p'*-TDE) (also known as DDD) | 3 | 58 | 10 | 21 | 185 |

[1] Current AL1 for DDT only are included in Table 5. There is no difference between the current AL and the AL1 German OSPAR review).

*3.7. Polybrominated Diphenyl Ethers (PBDEs)*

3.7.1. Revised Action Levels

There are no revised ALs for PBDEs or brominated flame retardants [7]. However, sample analyses of PBDEs are requested from areas in a limited number of applications that potentially have contamination.

3.7.2. Proposed Scenarios

There are no current action levels for PBDEs. A proposed scenario used FESG (Canadian Federal Environmental Sediment Guidelines [19] as AL2, and FESG divided by a factor of 3 for AL1 (based on a factor used for the derivation of AL1 for German values [8]). OSPAR [20] have corrected FESG [19] values to the standard 2.5% particulate organic carbon (POC) used for the other organic determinants assessed. The percentage number of samples below AL1 (FESG/3) and above AL2 (FESG) are presented (Table 19). The scenario showed varying results across the individual PBDEs, although most results in 100% of the samples fell below AL1. BDE100, BDE85 and BDE99 resulted in samples above AL2, with BDE99 (most common congener exceeded) resulting in 61% samples exceeding AL2. There were no data available for BDE209, but it is likely this will have exceedances in a number of places based on expert knowledge. The spatial extent of the data available is limited and is known to be focused on areas where PBDEs are expected but when mapped there may be regional variations in concentrations.

**Table 19.** Polybrominated diphenyl ether flame retardants (PBDEs)—Individual BDEs—FESG Percentage of samples below AL1 (FESG/3); above AL2 (FESG). n–No data available for BDE209. No FESG for BDE138 or BDE17.

| Polybrominated Diphenyl Ethers (PBDEs) | FESG/3 [19] AL1 (ppb) | Number of Samples below AL1 (%) | FESG [19] AL2 (ppb) | Number of Samples above AL2 (%) | Total Number of Samples |
|---|---|---|---|---|---|
| BDE100 | 0.3 | 44 | 1 | 14 | 141 |
| BDE153 | 367 | 100 | 1100 | 0 | 141 |
| BDE154 | 367 | 100 | 1100 | 0 | 141 |
| BDE183 | 4666 | 100 | 14,000 | 0 | 141 |
| BDE209 | 16 | n | 47.5 | n | 0 |
| BDE28 | 37 | 100 | 110 | 0 | 141 |
| BDE47 | 33 | 100 | 97.5 | 0 | 141 |
| BDE66 | 33 | 100 | 97.5 | 0 | 133 |
| BDE85 | 0.3 | 77 | 1 | 2 | 133 |
| BDE99 | 0.3 | 23 | 1 | 61 | 133 |

## 4. Discussion

This review has tested a variety of scenarios, comparing scenario results with current ALs where possible, to determine the potential impact of any changes that may be introduced in terms of sediment sample analysis. This review has not looked at the implications of each scenario in terms of the number of licenses that would be affected by the introduction of new ALs or the implications for the testing of sediment samples in terms of any lowering of AL1 which may require lower limits of detection, for example. The main conclusions for each contaminant type are indicated in the following sections.

### 4.1. Trace Metals

If the revised ALs [7] were wholly adopted this would mean an increase in AL1 for chromium and nickel and a reduction in AL1 for copper and mercury, as well as a decrease in AL2 for all metals. The range between AL1 and AL2 would be reduced for all metals. A minimal burden to ports/harbours can be expected if revised ALs are adopted. However, the 'High Level Review' [6] indicated that AL1s are conservative and therefore reducing these for copper and mercury goes against this recommendation. The increase in AL1 for chromium and nickel agrees with the recommendations from the 'High Level Review' [6]. A possible solution could be to maintain the current AL1 for copper and mercury but if an alignment with Scottish action levels is desirable, this will need agreement (as Scotland are already applying the revised ALs [7]).

The use of a regional metal AL1 concentration indicates a good potential to reduce the currently over conservative AL1s. In practice, the use of the port-derived background levels as defined [9] are proposed rather than the catchment-based averages used for scenario testing in this report. There are no regional background levels for Hg as no data were available in the datasets that could be used to derive the regional backgrounds [9]. The assessment of the regional action levels has only been carried out for England: if this approach was preferred and applied to the UK, additional work will be required to set regional ALs for the remaining Devolved Administrations.

### 4.2. Organotins

The use of the revised ALs [7] for organotins reduces the concentration permissible for disposal at the upper action level, which resulted in approximately 2% of samples exceeding AL2 for TBT but less than 1% for dibutyltin (DBT).

The assessment of the two novel scenarios, based on a reduced data set since the ban on TBT use (as areas where there are low results are no longer analysed), resulted in a greater number of sample results (~13%) exceeding AL2. The reduction in the concentration range between AL1 and AL2 means that there would be a reduction on the reliance of expert judgment in relation to the assessment of dredged material applications.

### 4.3. Polycyclic Aromatic Hydrocarbons (PAHs)

All scenarios tested for PAHs are likely to add some burden to ports and harbours as there are currently no ALs. The 'High Level review' [6] also recommends setting AL2 for any contaminants that do not already have these.

The proportion of samples with concentrations below the revised AL1 [7] for individual PAHs was variable, with all analytes showing an increase in the percentage falling below the AL1 if samples from Northumbria and the Humber are excluded. The exclusion of sample results from Northumbria and Humber was included for all scenarios to demonstrate this regional significance and the need for future consideration as to how the proposed ALs would be introduced.

Total Hydrocarbons (THC) are currently used in the absence of full data for PAHs to assess whether dredged material can be disposed of to sea by the regulators and their scientific advisors. However, the use of THC is limited as it provides no indication of toxicity, there is large inter-laboratory method variability and it appears to be conservative given that most sediments fail this threshold leading to additional evidence and assessments being required. Around 15% of samples had concentrations lower than the revised AL1 for THC, which increased to ~80% of samples when Northumbria and Humber were excluded.

The proportion of sample results below the Canadian threshold limit, ISQG (AL1) and above the PEL (AL2) for individual PAHs was variable depending on the individual PAH. The results showed that when data from Northumbria and the Humber were excluded, the percentage of sample results exceeding AL2 was reduced. For summed PAHs, ERLs and ERMs were the more sensitive scenarios tested when compared with $\sum$16PAH where very few sample results overall were above AL2. The results showed that $\sum$16PAH had a poor standard quality, because the analytes included in the sum were based on parent PAHs which have less toxicity than the alkylated PAHs measured, and because it shows that minimal sample results would be above AL2, potentially allowing more sediment for sea disposal than would be allowed when using other more sensitive approaches (e.g., the Gorham test using LMW/HMW [15]).

### 4.4. Polychlorinated Biphenyls (PCBs)

When applying the revised ALs [7] for PCBs, there was a minimal difference in the number of samples above AL2 for the $\Sigma$25_PCBs when compared with the number of sample results if the current AL2 was applied. The concentration range for the revised ALs was smaller than that for current ALs which would reduce the occurrence of reliance on expert judgement for assessing dredged material.

As there are no current ALs for individual PCBs, the scenarios exceeded the proposed AL2s. The 'High Level review' [6] recommends setting AL2 for any contaminants that do not already have these. The scenario based on the German National Action Levels [8] showed that the percentage of sample results falling below AL1 and exceeding AL2 was similar across the congeners although PCB52 showed a high proportion of sample results exceeding AL2. The scenario based on OSPAR EACs [17] shows variable results between the congeners. The results ranged from 98% of sample results falling below AL1 to 55% of sample results exceeding AL2. The application of the scenarios for the $\Sigma$25_PCBs and the $\Sigma$ICES7 based on the German OSPAR ALs [8] showed there was an increase in the

sample results exceeding AL2, but a higher proportion falling below AL1 compared to the current ALs.

### 4.5. Organochlorine Pesticides (OCPs)

There are only two current ALs for OCPs (dieldrin and dichlorodiphenyltrichloroethane (DDT)). There was an increase in the number of sample results exceeding AL2 for DDT and the proportions of sample results exceeding AL2 ranged from 0.9% for HCB up to 21.1% for TDE when the German OSPAR [8] ALs were applied to the test data. As there is no current AL for most of the OCPs, the application of ALs would allow an evidence-based assessment as opposed to a reliance on expert judgement and limitations given the small number of sites that are assessed for OCPs and the infrequency of these analyses. However, concentrations of OCPs are known to have reduced in the environment.

### 4.6. Polybrominated Diphenyl Ethers (PBDEs)

There are no current action levels for PBDEs. The scenario proposed based on FESGs [19] showed varying results across the individual PBDEs, although most resulted in 100% of the samples falling below AL1. BDE100, BDE85 and BDE99 resulted in samples above AL2, with BDE99 resulting in 61% of the samples exceeding AL2. As there are no current ALs, the application of ALs would allow an evidence-based assessment as opposed to a reliance on expert judgement which could be limited given the small number of sites that are analysed for PBDEs and the infrequency of these analyses.

### 4.7. Emerging Contaminants

About 120,000 chemicals are manufactured and imported into Europe. Many of these are known to be harmful to the aquatic environment and only a small fraction of these sites are monitored for the purpose of the disposal of dredge materials at sea. In actual fact, the current list of contaminants considered for a marine licence in the UK just cover a handful of 'legacy' contaminants. The latest addition, polybrominated flame retardants, were mostly banned from production and use in Europe from 2004 [21]. A wider range of hazardous chemical contaminants are monitored in the UK marine environment to meet the monitoring requirements of the OSPAR Coordinated Environmental Monitoring Programme (CEMP). However, in an overview of studies [22] that was performed to link chemical pollution in European river basins to measurable ecotoxic effects, it was concluded that the presence of contaminants listed as priority hazardous substances in EU rivers could only explain a small fraction of the toxicity observed.

Contaminants of emerging concern (CECs) or so-called 'emerging contaminants' is a generic term for a wide range and number of chemicals, with varying definitions. In Europe, these are generally defined as substances that have been detected in the environment that do not fall under regulatory surveillance programmes and whose fate and biological effects are poorly understood [23]. Examples of such compounds include pharmaceuticals, personal care products, pesticides, flame retardants and plasticisers, water repelling fluorinated chemicals, nanomaterials, microplastics, etc.

OSPAR has taken a systematic approach to identifying substances on the market that pose a risk to the marine environment based on either their persistence, liability to bioaccumulate and toxicity (PBT substances) or that give rise to an equivalent level of concern as the PBT substances. The resulting List of Substances of Possible Concern (LSPC) was adopted in 2002 and is revised regularly with 264 substances currently listed [24].

### 4.8. Implications

Except for the application of regional action levels, the amendment of action levels will have minimal impact on the UK's ability to report to LC/LP and OSPAR. It is worth noting that work for this project has already identified that there are differences in action levels being applied, both in terms of values as well as the fraction of sediment being measured. Further investigation of the implications of these differences is recommended to determine

how this impacts outcomes for LC/LP and OSPAR. The reports require action levels to be included but no direct assessment is undertaken by the UK, so data from previous years can be used. The application of regional action levels would potentially make future assessments on the condition of the marine environment more difficult without a change to the reporting format to link licence data to regional action levels. The assessment of regional action levels has only been carried out for England: if this approach was preferred and applied to the UK, additional work would be required to set regional action levels for the remaining Devolved Administrations.

The implementation of action levels for determinants which do not currently exist, in addition to the refinement of the current action levels will not only help the UK meet its international obligations, it will also help national policy ambitions to be met. It will help to achieve the UK High Level Marine Objectives by allowing assessments to be carried out on the material before it is disposed on the risk it poses, in terms of chemical contamination. It allows the standardisation of such assessments which supports the 25 Year Environment Plan and the UK's Marine Strategy to adopt higher standards of environmental protection through the reduction in AL2 (where appropriate and supported by evidence) and the reduction in the range between the two action levels, reducing the reliance on expert judgement, whilst increasing the ability to increase standardization of assessments.

In summary, Table 20 details the proposed action levels for each determinant.

**Table 20.** Proposed action levels for all determinants tested, including the current action levels for comparison.

| Contaminant Group—Units | Contaminant | Current AL1 | Proposed AL1 | Current AL2 | Proposed AL2 |
|---|---|---|---|---|---|
| Trace Metals –ppm | Arsenic (As) | 20 | 20 | 100 | 70 |
| | Cadmium (Cd) | 0.4 | 0.4 | 5 | 4 |
| | Chromium (Cr) | 40 | 50 | 400 | 370 |
| | Copper (Cu) | 40 | 30 | 400 | 300 |
| | Mercury (Hg) | 0.3 | 0.25 | 3 | 1.5 |
| | Nickel (Ni) | 20 | 30 | 200 | 150 |
| | Lead (Pb) | 50 | 50 | 500 | 400 |
| | Zinc (Zn) | 130 | 130 | 800 | 600 |
| Organotins —ppm | Dibutyltin (DBT) | 0.1 | 0.1 | 1 | 0.5 |
| | Tributyltin (TBT) | 0.1 | 0.1 | 1 | 0.5 |
| Polycyclic Aromatic Hydrocarbons (PAHs) —ppb | LMW [1] | | 552 | | 3160 |
| | HMW [1] | | 1700 | | 9600 |
| Polychlorinated biphenyls (PCBs) —ppb | Σ25_PCBs [2] | 20 | 20 | 200 | 180 |
| | ΣICES7_PCBs [2] | 10 | 10 | | 90 |
| | PCB28 | | 0.6 | | 1.7 |
| | PCB52 | | 0.9 | | 2.7 |
| | PCB101 | | 1 | | 3 |
| | PCB118 | | 0.2 | | 0.6 |
| | PCB138 | | 2.6 | | 7.9 |
| | PCB153 | | 13 | | 40 |
| | PCB180 | | 4 | | 12 |
| Organo-chlorine pesticides (OCPs) —ppb | Dichlorodiphenyltrichloroethane ($p,p'$-DDT) | 1 | 1 | | |
| | Dieldren | 5 | 5 | | |
| Polybrominated diphenyl ethers (PBDEs) —ppb | BDE28 | | 38 | | 110 |
| | BDE47 | | 33 | | 97.5 |
| | BDE66 | | 33 | | 97.5 |
| | BDE85 | | 0.3 | | 1 |
| | BDE99 | | 0.3 | | 1 |
| | BDE100 | | 0.3 | | 1 |
| | BDE153 | | 367 | | 1100 |
| | BDE154 | | 367 | | 1100 |
| | BDE183 | | 4666 | | 14,000 |
| | BDE209 | | 16 | | 47.5 |

[1] PAHs—LMW and HMW are defined in Table 12. [2] Σ25_PCBs and ΣICES7 are defined in Table 5.

The recommendations of this study are:

- Trace Metals. Consider adopting the revised ALs [7] and introducing background-based port AL1s. Application of background-based port AL1s beyond England would need to be considered by the devolved administrations. While the revised AL2s address the concerns raised in the 'High Level Review' [6] to some extent, they will still be either higher or equally the highest by comparison with other OSPAR countries with comparable AL2s (i.e., those analysing the <2 mm sediment fraction). Consequently, it would be prudent to review the trace metal AL2s further in due course. In addition, it would also be prudent to review the ALs where the potential inputs of certain trace metals are linked to emerging concerns (e.g., nanoparticles).
- Organotins. Consider adopting the revised ALs [7].
- Polycyclic Aromatic Hydrocarbons (PAHs). There are no current ALs, and all scenarios will therefore have the potential to cause additional burdens. The adoption of summed PAHs for LMW and HMW should be considered and new thresholds should be based on ERL/ERMs [15]. The introduction of the proposed ALs is likely to require either a phased approach where the implementation of proposed ALs is staggered to allow ports time to adjust to the management of dredging or a regional approach, particularly in Northumbria and the Humber. THC is highly limited and should not be proposed as a future AL.
- Polychlorinated biphenyls (PCBs). Revised ALs [7] for $\Sigma25\_PCBs$ and the phasing in of individual PCB ALs should be considered, as concentrations are still relatively high in the marine environment due to the persistent nature of these contaminants.
- Organochlorine Pesticides (OCPs). There are only two current ALs for OCPs (Dieldrin and dichlorodiphenyltrichloroethane (DDT)). No revised ALs. No changes to the current ALs proposed are generally known to have been reduced in the environment.
- Polybrominated diphenyl ethers (PBDEs). There are no current ALs for PBDEs. Minimal data are available. PBDEs are known to be a concern in the marine environment. Proposed ALs are based on FESG (Canadian Federal Environmental Sediment Guidelines as used for OSPAR MIME assessments) [19]. PBDEs ALs with analyses only should be requested if flagged as a high-risk area/ known area of concern or in relation to an incident. It is proposed that a baseline study would be useful to make sure any high-risk areas are not being missed.
- To maintain and update the collated dataset to help understand and sense check new applications.
- Other factors for future consideration highlighted include:
- To include water quality criteria.
- To keep abreast of new methods in assessing the environment, for example, passive sampling, which may become more dominant than direct testing and ensure the use of new methods and ALs are compatible.
- To identify high risk areas for emerging contaminants, including plastics, in dredge areas around the UK.
- To consider incorporating bioassays and ecotoxicology into the assessment framework and providing specific guidance on when it should be used, as currently UK use is limited.
- To consider introducing requirements for the measurements of black carbon and particulate organic carbon analyses in support of the interpretation of polycyclic aromatic hydrocarbons (PAHs).
- To improve understanding on how measuring uncertainty as a result of a wider range of laboratories as well as some difference in methodologies are now providing data for assessments since the 'High Level Review' [6] are now providing data for assessments.

## 5. Conclusions

This review has tested a variety of scenarios, comparing results with current ALs where possible, to determine the potential impact of any changes that may be introduced. Whilst most revisions would result in some samples falling into new categories i.e., below AL1, between AL1 and AL2 or above AL2, most of the changes would be minimal. The areas where the most change would be seen, unsurprisingly, are related to those determinants where there are currently no or only one AL. However, whilst the introduction of ALs where currently there are none would result in some samples being deemed unsuitable for disposal to sea, the assessment and decisions would be based on the best available evidence that is internationally accepted and would reduce the reliance on expert judgement which can yield differing results. However, these scenarios are based on the current suite of determinants that are analysed, some more frequently than others, but with the development of new knowledge and technology, emerging contaminants should also be continually reviewed and if needed, included in the assessment process, which may warrant additional ALs to ensure decisions are based on robust evidence. This may include recommending further analytical method development to determine lower concentrations than are currently measurable.

## 6. Patents

This paper is a summary of the Defra Action Level Review Report [13].

**Author Contributions:** Conceptualization, C.M., A.G. and C.V.; methodology, A.G.; software, A.G.; validation, C.M. and J.-A.L.; formal analysis, L.W., C.H. and J.B.; investigation, L.W., C.H. and J.B.; resources, C.M.; data curation, A.G., J.-A.L. and C.M.; writing—original draft A.G., J.-A.L., C.M., L.W., C.H., J.B., D.S., P.B. and A.B.; writing—review and editing, J.-A.L. and C.M.; visualization, A.G., C.M. and J.-A.L.; supervision, P.B. and D.S.; project administration, C.M.; funding acquisition, C.M. All authors have read and agreed to the published version of the manuscript.

**Funding:** This research was funded by the Department for the Environment, Fisheries and Rural Affairs (DEFRA), project ME5226.

**Institutional Review Board Statement:** Not applicable.

**Informed Consent Statement:** Not applicable.

**Data Availability Statement:** Data supporting reported results were collated from freely available data (MCMS) or on request from Devolved administrations (DAs).

**Acknowledgments:** The authors wish to acknowledge the support given by analysts, including all those responsible for completing contaminant measurements used in this publication, as well as technical support from Mark Etherton, project management from Adrian Bonfield and Leah Winpenny, and leadership advice from Clare Leech and Rachael Clarke, and review from Defra colleagues including Martin Lilley, Vanessa Fairbank and Steve Morris.

**Conflicts of Interest:** The authors declare no conflict of interest.

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
