# Peer review of "Reviewing the UK’s Action Levels for the Management of Dredged Material"

_geosciences, doi:10.3390/geosciences12010003_

Round 1

Reviewer 1 Report

Overview:

This paper describes a critical review of the UK’s action levels for contaminants evaluated for decisions of dredged material disposal suitability. Overall, this paper is well written and provides an extensive review of the implications of adopting revised ALs as informed by analysis of percentages of samples exceeding AL level 1 and 2 based on historic data.

Revisions are suggested below to improve the overall quality of the article.

A number of instances of “Error! Reference source not found” are throughout the paper and made the review challenging to understand which citations were being referenced. (e.g., Line 171; Line 182)

Line 12: suggested adding (2021) parenthetically after “current day”

Line15: change “hydrocar-bons” to “hydrocarbons”

Line 15: AL1, AL2 have not been defined. Perhaps add: A1 (acceptable for disposal) and A2 (unacceptable for disposal)

Line 162: Suggest included equations for the calculations referenced in sections 2.4.1 and 2.4.2

Line 197: Sentence needs to be revised, confusing as written: “Note trace metal analysis routinely includes arsenic, a non-metal and has been included as such for this review.”

Table 5: Are units important in the contaminant group column? This table just shows % of above or below standards.

Table 6: Footnote can be moved to “trace metals” since all parameters are listed with footnote

Line 253: delete parenthesis  

Figure 6: Suggest removing dotted line where not applicable (revised is same as current). The reader tends to look for it and gets confused when its not there. Or alternatively – make it another color and overlay so you can see it’s the same.

Figure 6: Do error bars only apply to Background x3? Presumably the variance is the same since each 2x and 3x are just multipliers. The figure description implies just 3x. Please clarify.

Section 4.1: suggest more discussion of the implication of increased AL1 for Cr and Ni. What are the practical near-term consequences of adopting the new AL1 in this context.

Table 12 and 13: Suggest including definition (list of analytes) as a footnote. This should greatly reduce the text in the table and make it easier for the reader to follow.

Author Response

Thank you for your review.

Please see point-by-point response to your comments:

A number of instances of “Error! Reference source not found” are throughout the paper and made the review challenging to understand which citations were being referenced. (e.g., Line 171; Line 182) - checked and now resolved.

Line 12: suggested adding (2021) parenthetically after “current day” - "current day replaced with 2018.

Line15: change “hydrocar-bons” to “hydrocarbons” - completed.

Line 15: AL1, AL2 have not been defined. Perhaps add: A1 (acceptable for disposal) and A2 (unacceptable for disposal) - definitions added.

Line 162: Suggest included equations for the calculations referenced in sections 2.4.1 and 2.4.2 Equations added.

Line 197: Sentence needs to be revised, confusing as written: “Note trace metal analysis routinely includes arsenic, a non-metal and has been included as such for this review.” Sentences revised to make clearer.Trace metal analysis routinely includes arsenic, a non-metal. Arsenic is included with trace metals even though it is a non-metal in this review. 

Table 5: Are units important in the contaminant group column? This table just shows % of above or below standards. Different units are relevant for different contaminant groups. This is made clearer in Table header: Contaminant group- units for Current AL1, Current AL2 and Range

Table 6: Footnote can be moved to “trace metals” since all parameters are listed with footnote. Moved to "trace metals".

Line 253: delete parenthesis  Parenthesis and contents deleted.

Figure 6: Suggest removing dotted line where not applicable (revised is same as current). The reader tends to look for it and gets confused when its not there. Or alternatively – make it another color and overlay so you can see it’s the same. Removed dotted line where not needed. If  revised and current AL1 the same, then revised dashed line removed and relabelled to current and revised AL1.

Figure 6: Do error bars only apply to Background x3? Presumably the variance is the same since each 2x and 3x are just multipliers. The figure description implies just 3x. Please clarify. Error bars are the same so have clarified in Figure heading:  Error bars (95% confidence limits calculated by standard deviation X number of samples X 0.05) are indicated as a line on each stacked bar and apply to the whole bar.

Section 4.1: suggest more discussion of the implication of increased AL1 for Cr and Ni. What are the practical near-term consequences of adopting the new AL1 in this context. Sentence added to highlight this fits with High Level Review recommendations, as AL1 for metals is relatively conservative: The increase in AL1 for chromium and nickel agrees with the recommendations from the ‘High Level Review’[6]. 

Table 12 and 13: Suggest including definition (list of analytes) as a footnote. This should greatly reduce the text in the table and make it easier for the reader to follow. -Completed.

Reviewer 2 Report

Lines 163-168: Some references are needed;

Lines 182 and 188: “Error! Reference source not found.” messages should be replaced by the correct references. By the way, these messages appear in many places along the text. They should be corrected;

Table 3: This table should be better discussed in the text;

Author Response

Dear reviewer

Thanks for your comments. Please see a point-by-point response to these:

Lines 163-168: Some references are needed - These metrics were devised by the authors. Details of calculations have been added.

Lines 182 and 188: “Error! Reference source not found.” messages should be replaced by the correct references. By the way, these messages appear in many places along the text. They should be corrected; - now sorted out.

Table 3: This table should be better discussed in the text - equations added to explain calculations.

Thanks